# Feature Importance Measurement based on Decision Tree Sampling

**Chao Huang** [1]  **Diptesh Das** [1]  **Koji Tsuda** [1 2 3]

## Abstract

Random forest is effective for prediction tasks but the randomness of tree generation hinders interpretability in feature importance analysis. To address this, we proposed a SAT-based method for measuring feature importance in tree-based model. Our method has fewer parameters than random forest and provides higher interpretability and stability for the analysis in real-world problems.

## 1. Introduction

Interpretable machine learning (ML) models are paramount for their seamless integration in high stake decision making problems e.g., medical diagnosis (Lakkaraju et al., 2016; Nemati et al., 2018; Ahmad et al., 2018; Das et al., 2019; 2022a;b; Adadi & Berrada, 2020), criminal justice (Angelino et al., 2017; Wang et al., 2022; Liu et al., 2022), etc. In medical diagnosis, specially in computer assisted diagnosis (CAD), model accuracy is important, but it is equally important for the doctor and the patient to know the features used in CAD modeling (Goodman & Flaxman, 2017; Rudin, 2019). There have been several established feature selection (FS) algorithms in ML literature, namely LASSO (Tibshirani, 1996), marginal screening (MS) (Fan & Lv, 2008), orthogonal matching pursuit (OMP) (Pati et al., 1993), decision tree (DT) based, etc. Among them, DT-based FS have been widely studied due to their high interpretability (Quinlan, 1986; 2014; Breiman et al., 1984).

Constructing an accurate and a small size (hence, better interpretability) DT is a challenging problem, and has been an active area of research over last four decades. Most of the existing methods are ad hoc, and do not have explicit control over the size and accuracy of a DT. For e.g., there are greedy splitting-based (Quinlan, 1986; 2014; Breiman et al., 1984), Bayesian-based (Denison et al., 1998; Chipman et al., 1998; 2002; 2010; Letham et al., 2015), branch-and-bound methods (Angelino et al., 2017) for DT construction.

Random forest (RF)(Breiman, 2001) is a widely used ensemble decision tree FS method. While RF has shown improvements in prediction accuracy and mitigating overfitting risk, due to the heuristic algorithms of decision tree generation, it often faces challenges such as the preference for larger trees, lack of statistical interpretability, randomness in feature importance measurement due the influence of many parameters, etc.

To handle those challenges, we propose a DT-based FS methods where we allow the user (e.g., a domain expert) to explicitly control the size and accuracy of a DT. We leverage a Boolean satisfiability(SAT)(Biere et al., 2009) encoding of a DT and perform uniform sampling of the SAT space with user-specified accuracy and size.Our method is a tree ensemble FS that generates small-size and high-accuracy decision trees, and determines the feature importance based on its emergence probability (i.e., the probability of a feature appearing in the high accuracy space).

Through numerical experiments, we evaluated our proposed method using four real-world dataset. We demonstrated that our method is capable of producing comparable accuracy as RF, but with small size DTs. We also compared our encoding with existing SAT-based encoding and demonstrated that our encoding scheme generates less variables and computationally more efficient. As we can uniformly sample form a high accuracy space with a specific tree size, this will allow us to formulate a statistical hypothesis testing framework to judge the significance of selected feature in terms of $p$-value and confidence interval which we consider as a potential future work.

## 2. Method

### 2.1. Feature importance using DT sampling

To naively determine the feature importance using decision tree (DT) sampling, one can enumerate all possible DT and find all the DTs exceeding an accuracy threshold. However, this naive approach is computationally prohibitive as the DT search space grows exponentially with the size of DT,

[1]Graduate School of Frontier Sciences, The University of Tokyo, 5-1-5 Kashiwanoha, Kashiwa 277-8561, Japan [2]Center for Basic Research on Materials, National Institute for Materials Science, 1-1 Namiki, Tsukuba, Ibaraki 305-0044, Japan [3]RIKEN Center for Advanced Intelligence Project, 1-4-1 Nihonbashi, Chuo-ku, Tokyo 103-0027, Japan. Correspondence to: Koji Tsuda <tsuda@k.u-tokyo.ac.jp>.

*Workshop on Interpretable ML in Healthcare at International Conference on Machine Learning (ICML)*, Honolulu, Hawaii, USA. 2023. Copyright 2023 by the author(s).

see Table 4 in appendix for details. Therefore, we propose a SAT based encoding of DT that reduces the search space significantly and also improves the sampling efficiency. The flow diagram of our method is shown in Figure 1. First, we encode DTs with specific size (#node) and accuracy (threshold) as a SAT problem represented in conjunctive normal form(CNF). Once the SAT encoding of DTs is constructed, any solution that satisfies all the constraints in the CNF file can be decoded into a valid decision tree of specific size and accuracy. Then, we utilize SAT sampling to generate multiple decision trees and calculate feature importance (emergence probability) based on the sampling results.

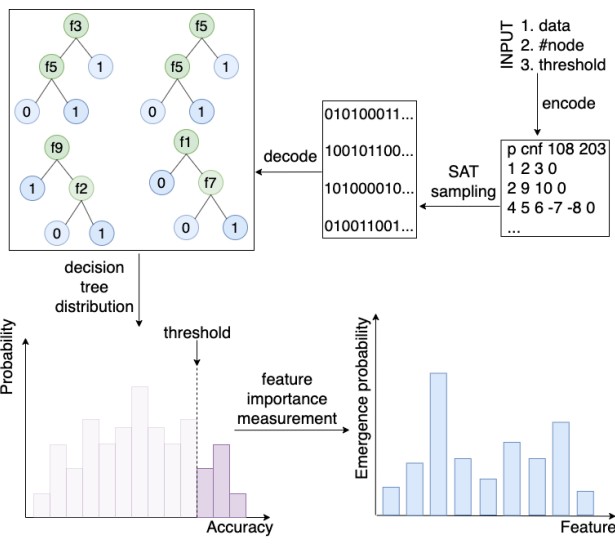

Figure 1. Feature importance measurement based on DT sampling

## 2.2. SAT-based DT encoding

Constructing decision trees with high accuracy and small size is an active area of research in the domain of constraint programming, and many Boolean satisfiability(SAT)-based encodings have been proposed in the literature (Bessiere et al., 2009; Narodytska et al., 2018; Verhaeghe et al., 2020; Janota & Morgado, 2020). To reduce the search space and enable fast sampling we proposed an efficient SAT-encoding of DT. Our encoding method is motivated by the method proposed in (Narodytska et al., 2018), but developed a new encoding (only branch node encoding) scheme that accelerates the process of SAT sampling significantly. We also introduced additional variables and constraints to make it possible to encode the DTs with any accuracy that you want. Encoding DTs with only branch nodes is non-trivial, the details of which have been provided below.

**SAT variables and constraints:** We consider the encoding of decision trees with 2N+1 nodes and the training

Table 1. Description of propositional variables

| Var | Description of variables |
|---|---|
| $vl_i$ | 1 iff branch node i has a left branch child, $i \in [N]$ |
| $vr_i$ | 1 iff branch node i has a right branch child, $i \in [N]$ |
| $l_{ij}$ | 1 iff node i has node j as the left child, with $j \in Child(i)$ |
| $r_{ij}$ | 1 iff node i has node j as the right child, with $j \in Child(i)$ |
| $lc_i$ | 1 iff class of the left leaf child of node i is 1, $i \in [N]$ |
| $rc_i$ | 1 iff class of the right leaf child of node i is 1, $i \in [N]$ |
| $a_{rj}$ | 1 iff feature $f_r$ is assigned to node j, $r \in [K], j \in [N]$ |
| $u_{rj}$ | 1 iff feature $f_r$ is being discriminated against by node j, $r \in [K], j \in [N]$ |

data consists of M samples and K features. Binary decision tree with 2N+1 nodes comprises N branch nodes and N+1 leaf nodes. The base method (Narodytska et al., 2018) sets the node ID sequentially as showed in Figure 2.a. Since it cannot distinguish between branch and leaf nodes by node IDs, branch and leaf nodes are assigned equivalent variables and are differentiated by additional constraints. In order to simplify it, we propose a method that only takes the branch nodes into consideration, viewing the leaf nodes as one of the properties of branch nodes as depicted in Figure 2.b.

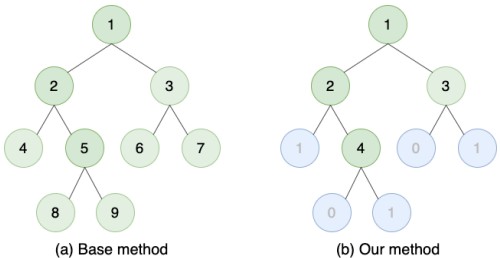

(a) Base method      (b) Our method

Figure 2. Node ID in SAT encoding

All the variables required to encode a DT are shown in Table 1, the subscript i and j represent the node ID (or index) while the subscript r denotes the feature ID(or index). For any natural number $n$, we use $[m:n] = \{m, m+1, \ldots, n-1, n\}$. The Function defined as $Child(i) = [i+1 : \min(2i+1, N)]$ can return possible node IDs of the children of $i^{th}$ node.

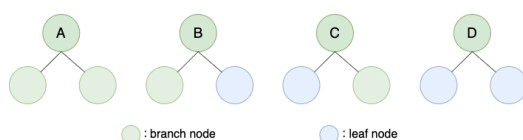

○ : branch node     ○ : leaf node

Figure 3. Four types of branch nodes

There are four types of branch nodes as depicted in Figure 3. We use $vl_i$ (resp.$vr_i$) variable to denote whether the $i^{th}$ node has a left (resp. right) branch child or not. With

$i \in [1:N]$ and $C \in \{0,1\}$:

$$vl_i = C \implies \sum_{j \in Child(i)} l_{ij} = C$$

$$vr_i = C \implies \sum_{j \in Child(i)} r_{ij} = C, \tag{1}$$

Every branch node (except root) has exactly one parent:

$$\sum_{i=\lfloor \frac{j}{2} \rfloor}^{j-1} (l_{ij} + r_{ij}) = 1, \ with \ j \in [2:N] \tag{2}$$

The IDs of branch nodes are assigned according to level order of the tree. For example, as shown in Figure 4, if $l_{35} = 1$, then $l_{26}$ or $r_{26}$ must be 0, because $6^{th}$ node cannot appear in front of $5^{th}$ node. With $i \in [1:N-1], j \in Child(i)$:

$$l_{ij} \vee r_{ij} \implies \sum_{h=1}^{i-1} \sum_{k=j}^{k=N} (l_{hk} + r_{hk}) = 0$$

$$r_{ij} \implies \sum_{k=j}^{k=N} l_{ik} + \sum_{k=j+1}^{k=N} r_{ik} = 0 \tag{3}$$

At any branch node, exactly one feature is assigned.

$$\sum_{r=1}^{K} a_{rj} = 1, \ with \ j \in [1:N] \tag{4}$$

Variable $u_{rj}$ has the information of whether the $r^{th}$ feature is discriminated at any node on the path from the root to this node. If the $r^{th}$ feature has already been assigned to one of ancestors, then it should not be assigned again. With $r \in [1:K], j \in [1:N]$:

$$\bigwedge_{i=\lfloor \frac{j}{2} \rfloor}^{j-1} (u_{ri} \wedge (l_{ij} \vee r_{ij}) \implies \neg a_{rj})$$

$$u_{rj} \iff (a_{rj} \vee \bigvee_{i=\lfloor \frac{i}{2} \rfloor}^{j-1} (u_{ri} \wedge (l_{ij} \vee r_{ij}))) \tag{5}$$

The encoding given by Formula (1)-(5) specify a space including all of valid decision trees of a given size but can't learn from the training data. To learn from the training data, we need to track if the $r^{th}$ feature was discriminated positively or negatively along the path from the root to $j^{th}$ node as proposed in (Narodytska et al., 2018). We adopted the same strategy in our method.

Furthermore, to sample decision trees with specific accuracy, we need to add an accuracy variable $w_t$, which is set

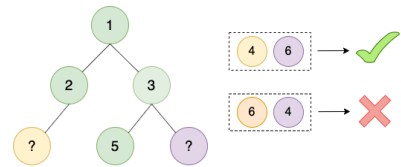

*Figure 4.* Sequential Node ID

to 1 if and only if the constraints required for correctly classifying the $i^{th}$ example are satisfied. Formula (6) is used to constrain the accuracy. For example, if the parameter $threshold = 80\%$, then the decision trees must correctly classify 80% samples at least.

$$\frac{1}{M} \sum_{t=1}^{M} w_t >= threshold, \ with \ threshold \in [0,1] \tag{6}$$

### 2.3. Decision Tree Sampling

**Sampling method:** To obtain samples from the decision tree space, we employ two SAT samplers: QuickSampler (Dutra et al., 2018) and UniGen3 (Soos et al., 2020). Quick-Sampler is a heuristic search algorithm that can generate large amounts of samples quickly. The algorithm starts with a random assignment and iteratively modifies the assignment by flipping the truth values of randomly selected variables. It is very efficient but the uniformity cannot be guaranteed. In contrast, UniGen3 is a more sophisticated algorithm for uniform SAT sampling with solid theoretical guarantees. It requires adding extra clauses to the encoding, which makes the sampling process computationally expensive.

**Sampling set:** Unigen3 and Quicksampler allow users to assign a subset of all the variables as sampling set. If the sampling set contains Y variables, the size of the solution search space will be $2^Y$. The samplers provide uniformity within the sampling set and increasing Y may adversely affect the sampling efficiency. Only part of valuables will be in the sampling set. For example, $u_{ij}$ is used to ensure that there are no repeated assigned features in any decision path but we don't need it during the decoding process. In addition, either the set $\{vl_i, vr_i\}$ or the set $\{l_{ij}, r_{ij}\}$ contains all the information needed to decode the tree structure, we only need to add one of them in the sampling set. Therefore, the smallest sampling set is $\{vl_i, vr_i, lc_i, rc_i, a_{rj}\}$.

**Feature importance measurement:** (Li et al., 2022) measured the importance of elements in sequences based on the distribution under a qualification threshold. Inspired by this concept, we define feature importance as the contribution of each feature to a high accuracy space. Specifically, within a space consisting of decision trees surpassing a given threshold, the contributions can be evaluated based on the

*Table 2.* Comparison of encoding size. $\#b, \#f$ denotes the number of training samples and the number of features respectively. $\#var$ denotes the number of variables used to build the encoding of the decision tree. $\#var\_cnf, \#cls\_cnf$ denotes the number of variables and the number of clauses in the Conjunctive Normal Form(CNF) file generated by tseitin transformation(Tseitin, 1983) provided in z3-solver(De Moura & Bjørner, 2008). $\#Ave.time$ denotes the time to generate 100 samples by unigen. We ran all the experiments (including the base encoding) on Intel(R) Xeon(R) CPU E3-1270 v6 @ 3.80GHz.

| Dataset | #b | #f | #n | thres. | #var | #var_cnf | #cls_cnf | Ave. time |
|---|---|---|---|---|---|---|---|---|
| mouse | 50 | 15 | 13 | 1.00 | 896/406 | 3599/1274 | 20586/8397 | 31.76/0.25 |
| | 50 | 15 | 11 | 0.90 | 747/336 | 3260/1990 | 22890/24689 | 1028.02/567.08 |
| car | 40 | 10 | 17 | 1.00 | 866/390 | 3956/1406 | 21625/9495 | 260.68/65.47 |
| | 50 | 15 | 11 | 0.90 | 747/336 | 3436/2152 | 26218/33473 | 1722.26/310.99 |
| breast | 40 | 15 | 17 | 1.00 | 1206/550 | 5821/2041 | 32400/13663 | 96.34/2.32 |
| | 50 | 12 | 13 | 0.90 | 740/334 | 3759/2198 | 24732/27869 | 407.32/125.81 |
| heart | 40 | 19 | 13 | 1.00 | 1104/502 | 4572/1590 | 26063/10697 | 79.87/11.62 |
| | 50 | 10 | 13 | 0.90 | 636/286 | 3241/2108 | 21568/25552 | 1671.18/327.89 |

probability of each feature appearing in this space (we name it as emergence probability). Since we sample decision trees from uniform distribution, we can estimate the probability by just counting how many times each feature appears. Random forest often uses feature permutation or mean decrease in impunity to calculate feature importance. It's also possible to apply these approaches to our framework.

# 3. Results

**Comparison between DT-sampler and RF:** We compared our method with RF on several real-world benchmark datasets(Dua & Graff, 2017). As shown in Table 3, our method can provide similar accuracy compared with random forest even if we only sample decision tree in small space. Relying on heuristic rules to build decision trees, random forest tends to generate larger decision trees. Besides, the randomness of tree generation makes it difficult to generate stable results for feature importance measurement (see Figure 7 in appendix for details). In the appendix (see Figure 8), we provide more detailed results on the stability of our method's feature importance measurements.

*Table 3.* Comparison of tree sizes and accuracy. Grid search on parameters = {'max_leaf_nodes':[3,6,9,12,15,18,21,None]} is utilized to run random forest and all the results are the average of three experiments on different subsets of the corresponding datasets, shown in the order of RF/ours. #b,#f denotes the number of training samples and the number of features respectively.

| Dataset | #b | #f | thres.(%) | #node | training acc.(%) | test acc.(%) |
|---|---|---|---|---|---|---|
| mouse | 50 | 15 | 92.0 | 6.90/7 | 98.00/96.00 | 91.67/93.33 |
| car | 100 | 15 | 92.0 | 16.93/11 | 98.00/93.00 | 88.57/88.32 |
| breast | 150 | 15 | 81.6 | 19.00/11 | 83.33/80.89 | 73.49/74.54 |
| heart | 170 | 19 | 81.0 | 23.00/15 | 89.21/84.50 | 83.93/81.36 |

**Comparison with existing SAT encodings:** Our new encoding of tree structure reduces a large part of variables and constraints compared to the encoding method in

(Narodytska et al., 2018). The results on several benchmark datasets proved the acceleration in the process of SAT sampling as shown in Table 2.

## 3.1. Interpretation

We define feature importance as its emergence probability in the high accuracy space as mentioned in 2.3. Parameter *threshold* is used to describe what a high accuracy space means and its value depends on specific real-world scenarios and the desired level of strictness regarding accuracy requirements. To demonstrate our method, we utilize decision tree sampling on a subset of the breast-cancer dataset, which consists of 150 samples and 15 selected features (refer to Figure 6 in appendix). Initially, we set the threshold to 0, allowing for the random sampling of any decision tree. In this case, each feature is assigned to any branch node with equal probability, resulting in an emergence probability of $\frac{1}{15}$ for each feature. However, as we increase the threshold, the emergence probabilities of the features differ. Features with an emergence probability $\geq \frac{1}{15}$ are considered important.

# 4. Conclusion

We proposed a method to measure feature importance using DT sampling and compared our method with random forest using four real-world datasets. Due to the randomness in tree generation and over-dependence on many parameters, RF-based feature selection generates unstable results. Our method provides a principled framework to measure feature importance based on sampling results from a high accuracy space with a clear threshold, which provides stable analysis results for real-world problems. Potential future research direction can be the development of a statistical hypothesis testing framework on top of our proposed DT sampling method to judge the reliability of feature selection and or the development of a fast SAT solver using quantum annealing or other QUBO-based solver.

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

# A. Appendix

## A.1. Ground Truth of sampling

Generating all decision trees of size 2N+1 can be done by following procedures.

1. generate all possible binary trees of size N. The number of binary trees of a given size is the Catalan number $C_N$, which is given by the formula: $C_N = \frac{(2N)!}{((N+1)! \times N!)}$.
2. Add N+1 leaf nodes to each tree and assign decision values to the leaf nodes. Since every leaf node can be 0 or 1, N+1 leaf nodes will have $2^{N+1}$ different combinations.
3. Assign a feature to each branch node. Note that we always obey the rule that there is no repeated features in any path from root to any leaf.

The ground truth of the sampling result is the distribution of all the decision trees in the whole space. We calculated the ground truth of the decision tree distribution when $\#node \leq 11$ on a subset of binarized car dataset with 10 features. As showed in Table 4, the computation time will increase rapidly along with the increment of size, which is also the reason why we need to do sampling instead of enumerating. Figure 5 shows the comparisons among the ground truth and the sampling results obtained by different samplers.

Table 4. The time to get all of decision trees. The time results of the two last rows are estimated and we didn't consider any multi-thread or multi-process strategies here.

| #nodes | #valid structures | #decision trees (Upper Bound) | Time |
|---|---|---|---|
| 3 | 1 | $1 \times 10^1$ | fast |
| 5 | 2 | $2 \times 2^3 \times 10^2$ | fast |
| 7 | 5 | $5 \times 2^4 \times 10^3$ | fast |
| 9 | 14 | $14 \times 2^5 \times 10^4$ | 365s |
| 11 | 42 | $42 \times 2^6 \times 10^5$ | 6h approx. |
| 13 | 132 | $132 \times 2^7 \times 10^6$ | 377h est. |
| 15 | 429 | $429 \times 2^8 \times 10^7$ | 24,505h est. |

## A.2. Interpretation

The results of feature importance change with the definition of high accuracy space. As shown in Figure 6, if we consider decision trees with an accuracy of at least 75% as effective, the important features would be $\{f_6, f_7, f_9, f_{10}, f_{11}, f_{15}\}$. On the other hand, if we raise the accuracy requirement to a minimum of 83% for each effective decision tree, the important features would change to $\{f_6, f_7, f_{10}, f_{11}, f_{15}\}$.

## A.3. Comparisons of Feature Importance between DT-sampler and RF

We did experiments on a subset with 100 samples of breast-cancer dataset to compare our method with random forest in feature importance measurement. Our method shows

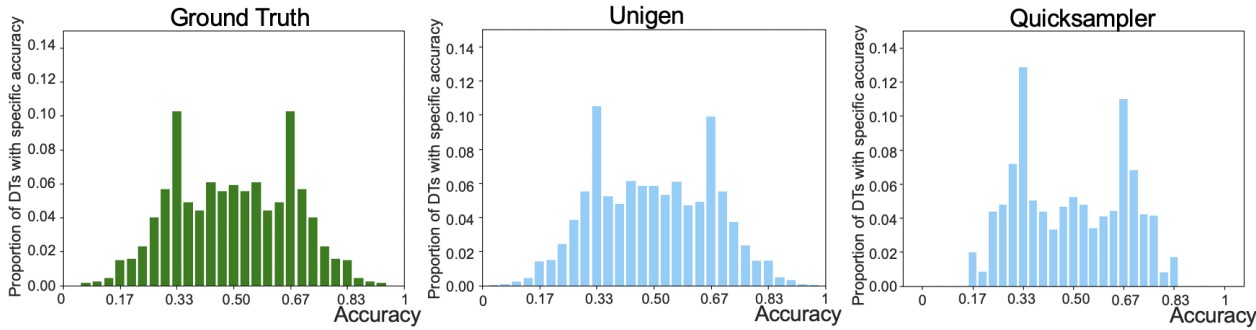

*Figure 5.* Comparisons of sampling effectiveness

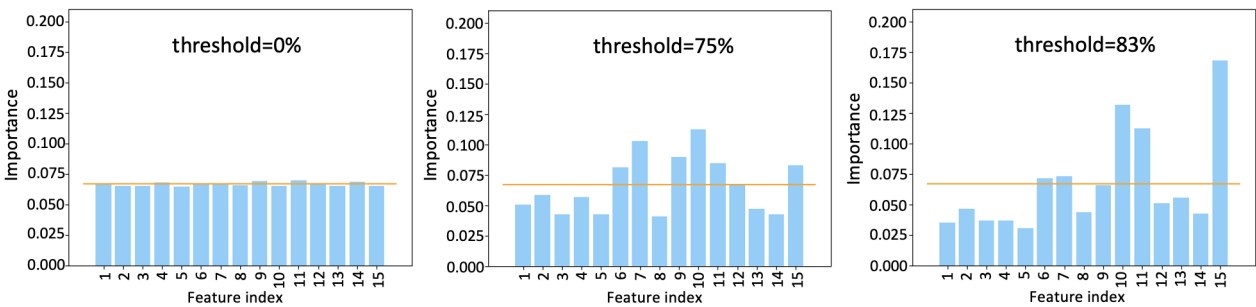

*Figure 6.* Feature importance results on breast dataset with different thresholds. Orange lines denote the emergence probability of features assigned completely randomly. The meaning of each feature: {f1: age, f2: menopause, f3-5: tumor-size, f6-8: inv-nodes, f9: node-caps, f10-12: deg-malig, f13: breast, f14: breast-quad, f15: irradiat}

superior stability compared to Random Forest. In Figure 7, we observe that when different random seeds or parameters are used, the distribution of decision trees generated by Random Forest consistently changes. This variability in tree generation directly impacts the feature importance results, leading to significant differences.

Furthermore, Random Forest tends to generate a large number of trees with low accuracy, making it unreliable to measure feature importance for real-world problems. In contrast, our method calculates feature importance based on decision trees sampled exclusively from a high accuracy space, which ensures the stability and interpretability of our results as depicted in Figure 8.

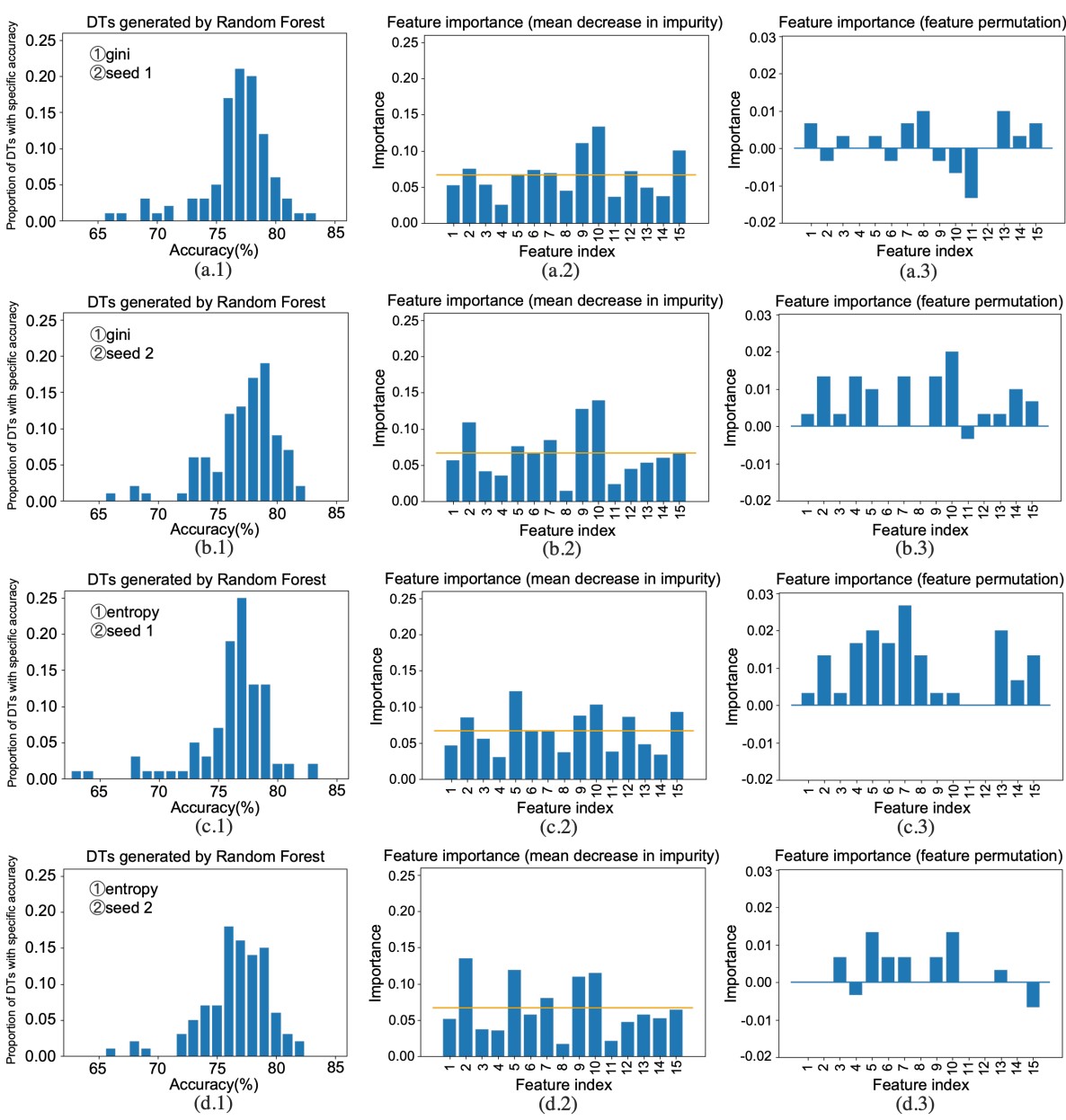

*Figure 7.* Drawbacks of random forest. The four rows of figures show the results of three experiments on breast dataset using different random seeds and splitting criterion as random forest parameters. The first column shows the training accuracy distribution of the decision trees generated by random forest. The second and third columns show the feature importance measured by mean decrease in impurity and feature permutation respectively.

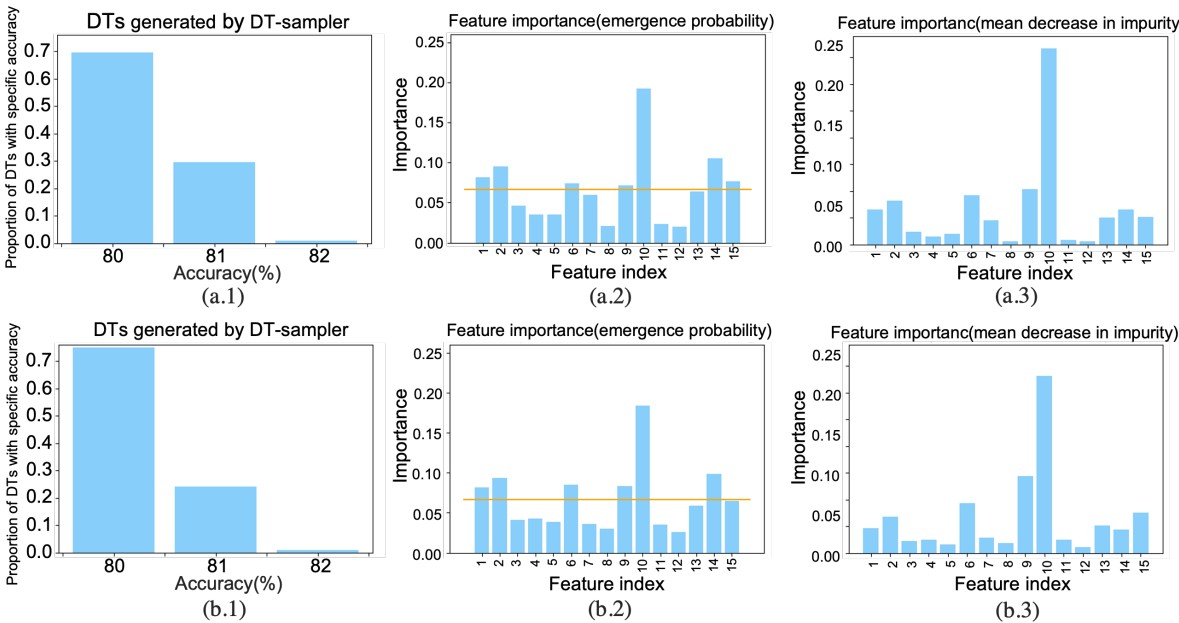

*Figure 8.* Stability of decision tree sampling. The two rows of figures show the results of two experiments on breast dataset using different random seeds during decision tree sampling. The first column shows the training accuracy distribution of the sampling results. The second and third columns show the feature importance measured by emergence probability and mean decrease in impurity respectively.

