# OpenReview forum: "Feature Importance Measurement based on Decision Tree Sampling"
_ICML.cc/2023/Workshop/IMLH — IMLH 2023 PosterShortPaper_

### Official Review · Reviewer_bikS · 2023-06-06
**Extension of random forest feature importance with SAT encoding, enhancing stability and interpretability, but can improve in baseline comparisons and consistency of shown results.**

**Rating:** 6
**Confidence:** 3

**Review:**

### Summary of the Paper (Short Paper):
The paper presents an extension to classical feature importance from random forest models which leverages SAT encoding based sampling for more stability and interpretability. It shows the effectiveness of the approach on four datasets containing tabular data.

### Quality:
Explanation of the method and the significance of the problem are both well motivated. The paper shows the effectiveness of the approach through higher test set accuracy (Table 3) and better feature importance stability compared to two other approaches (mean decrease and feature permutation, Figure 7). The paper would however benefit if popular model-agnostic methods such as SHAP [2] or LIME [3] based feature importance or comparable SAT-based methods as the mentioned work from Narodytska et al. [1] would be added for comparison.

While the paper shows the result on four datasets in Table 3, only three datasets are shown in Table 2 and Figure 6,7, and 8 only show results for one dataset. A consistency to always show the results for all included datasets would benefit the paper.

### Clarity:
The paper is very well written. Especially the abstract is short and crisp. Figures help to understand the presented concepts.

Minor remarks:
- Abbreviation not introduced: MS (line 32), OMP (line 33)

### Originality:
The paper acknowledges the similarity to the approach of Narodytska et al. [1] , however only argues that operational constraints like number of variables and computation time are solved more efficiently by their method. Further, Narodytska et al. seem to use ab less powerful CPU (Intel(R) Xeon 3.50GHz) and different Solvers (e.g. Glucose3) with memory and time limits. I can not comment on how these decisions affect the computation time. It would benefit the significance if the paper also commented on performance and faithfulness and stability of the feature importance between both approaches.

### Significance:
Feature Importance of classification and regression trees is still one of the most used interpretability of machine learning models. While the presented adoption could be argued as small, the widespread use of feature importance adds to the significance.

[1]: Narodytska, N., Ignatiev, A., Pereira, F., Marques-Silva, J., & Ras, I. (2018, July). Learning Optimal Decision Trees with SAT. In Ijcai (pp. 1362-1368).

[2]: Lundberg, S. M., & Lee, S. I. (2017). A unified approach to interpreting model predictions. Advances in neural information processing systems, 30.

[3]: Ribeiro, M. T., Singh, S., & Guestrin, C. (2016, August). " Why should i trust you?" Explaining the predictions of any classifier. In Proceedings of the 22nd ACM SIGKDD international conference on knowledge discovery and data mining (pp. 1135-1144).

---

### Official Review · Reviewer_uUE2 · 2023-06-17

**Rating:** 6
**Confidence:** 3

**Review:**

This paper introduces an innovative method for measuring feature importance in tree-based models, aiming to enhance their interpretability. The authors propose a SAT-based approach that simplifies the analysis while providing valuable insights. By encoding decision trees as SAT problems, the method reduces the complexity of searching for important features and improves sampling efficiency. This enables the generation of decision trees with specific sizes and accuracies, allowing users to have explicit control over the interpretability and accuracy of the model.

The authors' approach shows promise in producing smaller-sized decision trees with comparable accuracy to random forests. The proposed method's simplicity and higher interpretability make it appealing for various real-world problems, such as medical diagnosis and criminal justice. By highlighting the emergence probability of features based on sampling results, the method offers insights into their relative importance. While further empirical validation and a more comprehensive comparison with existing methods would strengthen the study, this paper presents a valuable contribution toward understanding feature importance in tree-based models.

---

### Meta-Review · Area_Chair_bXvk · 2023-06-18

**Recommendation:** Accept (Poster)
**Confidence:** 4

**Metareview:**

An extension to classical feature importance from random forest models is presented in the paper. This extension leverages SAT encoding based sampling to offer more stability and interpretability. The effectiveness of this approach is demonstrated on four datasets containing tabular data.

The paper has been deemed interesting by all reviewers. Some concerns have been raised regarding references and consistency of the results. The authors are encouraged to address these weaknesses in the next revision, as outlined in the reviews.

---

### Decision · Program_Chairs · 2023-06-20

Accept (Poster Short Paper)